# AFM Characterization of Track-Etched Membranes: Pores Parameters Distribution and Disorder Factor

**Alina V. Golovanova** [1,2,3,*], **Mariia A. Domnina** [1,2,4], **Artem I. Arzhanov** [1,2,3], **Kamil R. Karimullin** [1,2,3], **Ivan Yu. Eremchev** [1,2,3] and **Andrey V. Naumov** [1,2,3]

1  Institute of Spectroscopy of the Russian Academy of Sciences, 108840 Moscow, Russia; mariia_domnina@mail.ru (M.A.D.); arzhanov.artyom@gmail.com (A.I.A.); kamil_karimullin@mail.ru (K.R.K.); eremchev@isan.troitsk.ru (I.Y.E.); a_v_naumov@mail.ru (A.V.N.)
2  Laboratory of Physics of Advanced Materials and Nanostructures, Moscow State Pedagogical University, 119435 Moscow, Russia
3  Lebedev Physical Institute of the Russian Academy of Sciences, Branch in Troitsk, 108840 Moscow, Russia
4  Moscow Institute of Physics and Technology, 141700 Dolgoprudny, Russia
*  Correspondence: golovanova@isan.troitsk.ru

**Abstract:** The structural characteristics of polymer track-etched membranes (TM) were obtained by atomic force microscopy (AFM) for a set of samples (polypropylene, polycarbonate, polyethylene terephthalate, with average pore diameters ~183, 375, and 1430 nm, respectively). The analysis of AFM experimental data was performed by using a specially developed technique for computer analysis of AFM images. The method allows one to obtain such parameters of TM as distribution of pore diameters, distribution of the minimum distances between the nearest pores, pore surface density, as well as to identify defective pores. Spatial inhomogeneities in the distribution of pore parameters were revealed. No anisotropy (some specific selected direction) was found in the surface distribution of the pores in the samples under study.

**Keywords:** track-etched membranes; atomic force microscopy; image analysis; pore parameters

## 1. Introduction

Track-etched membranes (TM) are promising materials compared with conventional membranes due to their well-defined structure. TM and the composite materials based on them are key elements for many applications [1]. They are widely used as model systems and templates for the synthesis of micro- and nanostructures [2,3] and composites (metalized, hybrid, magnetoactive, electroactive, e.g., SERS-active substrates [4,5]), in technological and laboratory filtration [6], cell cultivation [7], etc. Among all areas, the use of TM in biology is especially important [8,9].

The size and shape of pores can be changed in a controlled manner to obtain a membrane with the desired transport and retention characteristics [10,11]. Since track membranes are manufactured by using heavy-ion accelerators, the density and angular distribution of pores can be easily varied depending on the ion flux parameters. The pore shape can be cylindrical, conical, funnel-like, or cigar-like, depending on the etching parameters. To date, the process of TM formation has been studied in detail. TM are made of polymeric materials such as polyethylene terephthalate (PETP) or polycarbonate (PC). TM have many advantages such as pore identity, the possibility of a wide variation in pore density (depending on the mode of irradiation), the ability to change the direction of the pores, the possibility to change the shape of the pores, low price in comparison with their analogs. However, the disadvantages of TM are the disorder in pore distribution, their low heat resistance, the relatively high roughness of the pores' walls (as compared with matrices of porous alumina).

During TM synthesis, variations in the parameters of ion beams and chemical etching can lead to a random distribution of the pores in the sample. Recently, new methods of

TM synthesis leading to an ordered distribution of pores have been proposed [12,13]. For the synthesis of such membranes, the synchrotron radiation is structurally ordered using a system of multi-beam lattice interference lithography. The synthesis is carried out in a hydrogen chamber. Hydrogen reacts photochemically with a polymer film on irradiated sections, with the formation of volatile products, which are then removed. As a result, one can obtain a membrane filter with specified parameter and ordered orientation, size, and shape of the pores.

Due to the growing interest in track membranes, it is required to develop a new method for their structural characterization such as for the determination of the distribution of pore size and of the distances between the pores. It is also necessary to evaluate new parameters that characterize the order (disorder) in the orientation of the pores. An important parameter that can affect the functional properties of a porous material is the fluctuation of pore density in the volume of the sample.

One of the promising methods of TM characterization is fluorescence microscopy [14–16] using organic luminophores or quantum dots as spectral probes [17,18]. Usually, TM are characterized by electron microscopy [19–22] with spatial resolution up to angstroms, but this method is expensive and requires a complex procedure in sample preparation. Moreover, electron microscopy assumes the presence of a vacuum, which makes it impossible to study biological objects in vivo. Therefore, only frozen, dried, i.e., non-vital, objects can be investigated by this method.

Along with electron microscopy, in such studies, the technique of atomic force microscopy (AFM) can be used [3,23], which is more affordable financially and does not require additional sample preparation. The AFM method has a high spatial resolution and the ability to implement non-destructive diagnostics of the samples, which is especially important for the study of biological objects under normal conditions, also in combination with the Raman spectroscopy technique. The AFM technique can be used to obtain information on the main stages of synthesis and to study the structure of TM. One more practical motivation to develop a technique for the characterization of spatial distributions and of the distribution of pores parameters in TM is their application in the template synthesis of metal nanostructures in TM pores. The properties of metal nanowires (as well as of more complicated secondary nanostructures, such as tubes, dendrites, etc.) and their further agglomeration, as well as mechanical and electrodynamic properties, depend on pores geometrical parameters and their distributions. This defines the functional properties of metasurfaces prepared in such a way, e.g., the enhancement coefficient for SERS measurements [4,5,24].

Here, we applied the AFM technique to characterize polycarbonate (PC), polyethylene terephthalate (PETP), and polypropylene (PP) TM; we used AFM image processing and statistical methods to analyze the distribution of TM pores parameters.

## 2. Samples and Experimental Technique

The objects of our study were TMs obtained with the following procedure. Polymer films "Torayfan", T2372 (Toray, Japan) with a thickness of 10 μm were irradiated with accelerated Xe ions with energy of 125 MeV using the U-300 accelerator at the Flerov Laboratory of Nuclear Reactions of the Joint Institute for Nuclear Research (JINR, Dubna, Russia). Next, the films were treated with a solution of chromic anhydride (250 g/L) in 40% sulfuric acid at a temperature of 80 °C to form pores of the required diameter. The following samples were taken for the study: polycarbonate (PC) with pores of 400 nm in diameter, polyethylene terephthalate (PETP) with pores of 1 μm, and polypropylene (PP) with pores of 240 nm (pore sizes were indicated by the manufacturer). These samples were synthesized by Prof. P. Apel (JINR) and kindly presented in our joint work previously [15].

The atomic force microscope NTEGRA PRIMA (NT-MDT) was used to obtain topography maps of the samples. This microscope makes it possible to obtain images at different scales depending on which auxiliary equipment is used. It is equipped with an optical viewing system, as well as with the isolated table TableStable TS-150 which provides active

vibration protection (within the range from 0.7 to 1000 Hz). For accurate measurements, we used the SMENA SFC050PRO measuring head, which allows obtaining topography maps with an extremely low noise level in Z coordinate (RMS = 0.05 nm) within the area of 100 × 100 μm. Before measuring each sample, a calibration was performed using the TGZ-2 periodic structure.

Silicon cantilevers NSG-10 (NT-MDT) with a typical radius of curvature of 6–10 nm were used as a probe. To obtain a topography map of the samples, all measurements were carried out in the semicontact (tapping) mode, in which the cantilever tip slightly touches the sample surface at the lowest point of its oscillations. This mode was used to minimize the deformation of the sample due to the friction of the cantilever on the sample surface during scanning, since the needle only "tapped" the sample.

## 3. Instrumental Response Function of the AFM

One of the key parameters characterizing the experimental setup of the AFM is the instrumental function of the probe. It directly affects the quality of the obtained topography map of the sample, for example, for a single pore of a TM (Figure 1a).

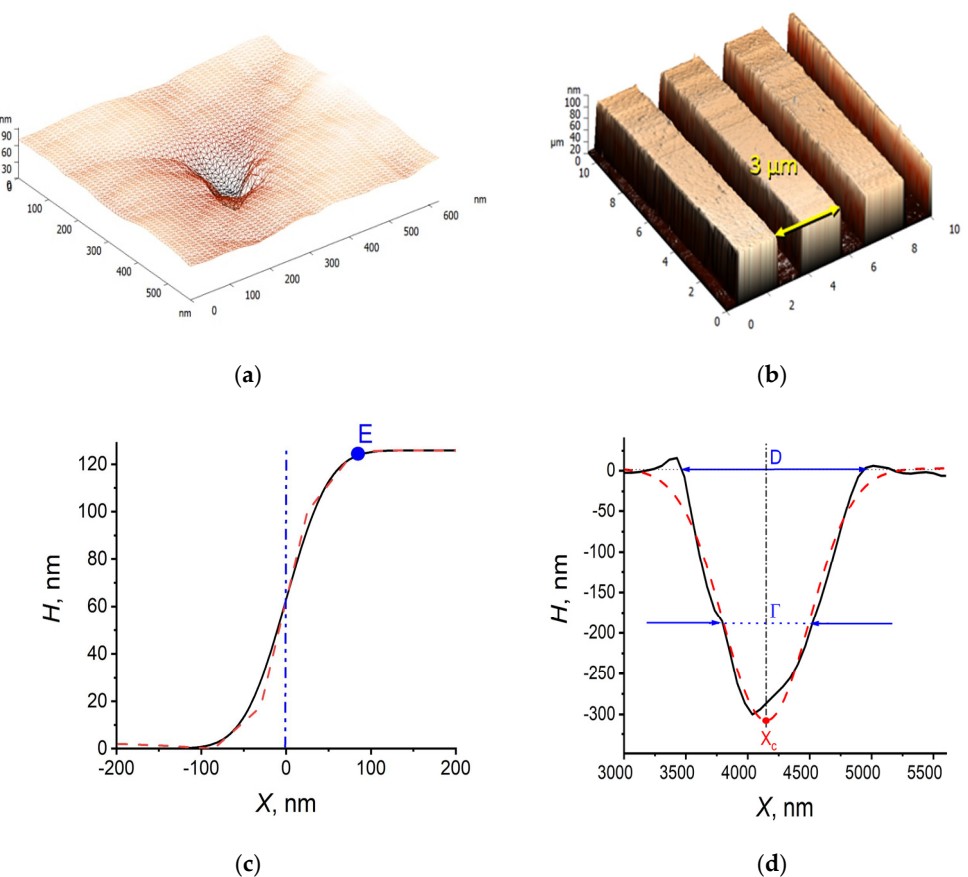

**Figure 1.** (**a**) AFM image (topography map) of a single TM pore. (**b**) AFM image of the calibration periodic structure TGZ2. (**c**) Measured profile of a single step item in the TGZ2 structure (red dashed line) and the result of modeling this profile using the specified instrumental function (see Equation (1)) of the AFM (black solid line). (**d**) Single pore profile measured in PETP TM (black solid line) and the result of its approximation by the Gaussian function (green dash-dotted line).

To determine the instrumental response function, the calibration silicon periodic structure TGZ2 was measured, which is a grating with a period of 3 μm and a step height of 110 nm (see Figure 1b).

The measured profile of a single step item in the TGZ2 structure (Figure 1c, black solid line) was fitted with the convolution of Heaviside $H(x)$ and Gaussian functions $G(x)$:

$$F(x) = C \cdot \int_{-\infty}^{\infty} H(x') \cdot G(x - x') dx' \tag{1}$$

Here, $x$ is the coordinate along the cut of the sample surface. The Heaviside function, which describes the profile of the real single step item in the TGZ2 structure, was used as a model. The Gaussian function was selected as an approximation of the instrumental response function of the AFM probe:

$$H(x) = \begin{vmatrix} 1 \; if \; x \geq 0 \\ 0 \; if \; x < 0 \end{vmatrix} \; and \; G(x) = \frac{1}{\gamma\sqrt{2\pi}} e^{\frac{-(x-m)^2}{2\gamma^2}} \tag{2}$$

The fitting result using expression (1) of the experimentally measured TGZ2 profile structure (red dashed line in Figure 1c) is shown in Figure 1c by the black solidline. The value of the width parameter of the Gauss function $\gamma$ was found to be equal to 40 nm. Thus, the tilt of the measured curve can be characterized by the value of full width at half maximum of the fitting curve, which was equal to 40 nm (Figure 1c) for the TGZ2 profile step with the depth of 100 nm. This value represents the effective width of the AFM tip at some height; the deeper the hole we characterize, the larger the effective width is. Thus, if a narrow hole is profiled, a smoothed peak shape will be found in the experiment (Figure 1a). At the same time, theposition of the pore edge (point E on Figure 1c) can be found with extremely high precision (higher than 5 nm).

To simplify and automate the procedure of pore recognition in AFM images and pore characterization, a two-dimensional Gaussian function (2D-Gauss) can be used to describe the profile of a pore, so the AFM profile of each pore is fitted by 2D-Gauss:

$$f(x,y) = f_0 + \frac{A}{\Gamma} \; \exp\left(-\frac{(x - X_c)^2}{\Gamma^2}\right) \exp\left(-\frac{(y - Y_c)^2}{\Gamma^2}\right), \tag{3}$$

Here, $f_0$ is the base, $A/\Gamma$ is the depth of the measured pore profile, $2\Gamma$ is the width of the measured pore profile at $1/e$ height, $X_c$ and $Y_c$ are the lateral coordinates of the geometrical center of the pore. This original software has been developed and used for the analysis of fluorescence images of single molecules [25]. An example of the approximation of a pore profile measured in PETP TM by the one-dimensional Gaussian function is shown in Figure 1d.

Further, it was necessary to find the coefficient for recalculating the width of the measured pore profile $\Gamma$ with the true value of pore diameter $D$ (see Figure 1d). After the analysis of the data obtained for pores with true diameters in the range of 183 to 1430 nm, we found this coefficient as high as 4.2 ($\pm$5%); thus, for each pore, we determined the diameter as $D = 4.2\Gamma$. The parameters $X_c$, $Y_c$, and $D$ were further analyzed.

## 4. Characterization Parameters and Model Calculations

In order to characterize TM, we suggest using the following parameters and their distributions:

- pore diameters $D$ and their distribution $P_D$;
- the relative number of somehow defected pores $\eta_{defect}$;
- distances between the nearest pores $R_{min}$ and their distribution $P_{Rmin}$;
- pore density $\rho$ (one can also characterize the spatial inhomogeneity of pores distribution by finding the distribution $P_\rho$ of pores densities in different regions of the sample, i.e., search for density fluctuations);
- presence/absence of "anisotropy" of pores distribution over a surface.

The number of defective pores was calculated in the following way. The pore was considered defective if the approximate parameter $D$ exceeded by more than $2\sigma$ the average size $\langle D \rangle$ of all pores (here $\sigma$ is the standard deviation of $P_D$). Two adjacent pores were considered defective pores also if the distance between their centers $R_{min}$ was comparable to their sizes $D$.

To determine the presence of "anisotropy" of pores distribution over a surface, we investigated the distribution of the angles $\varphi$ of the segments connecting the centers of the nearest pores with respect to the direction specified for the sample used. The histogram of the distribution $P\varphi$ (some histogram show the orientations of the nearest pores pairs) was plotted in the polar coordinate system (see Figure 2g–i). Each petal in such a ring histogram represents the relative number of nearest pores pair, which are oriented with respect to each other with a certain angle bin.

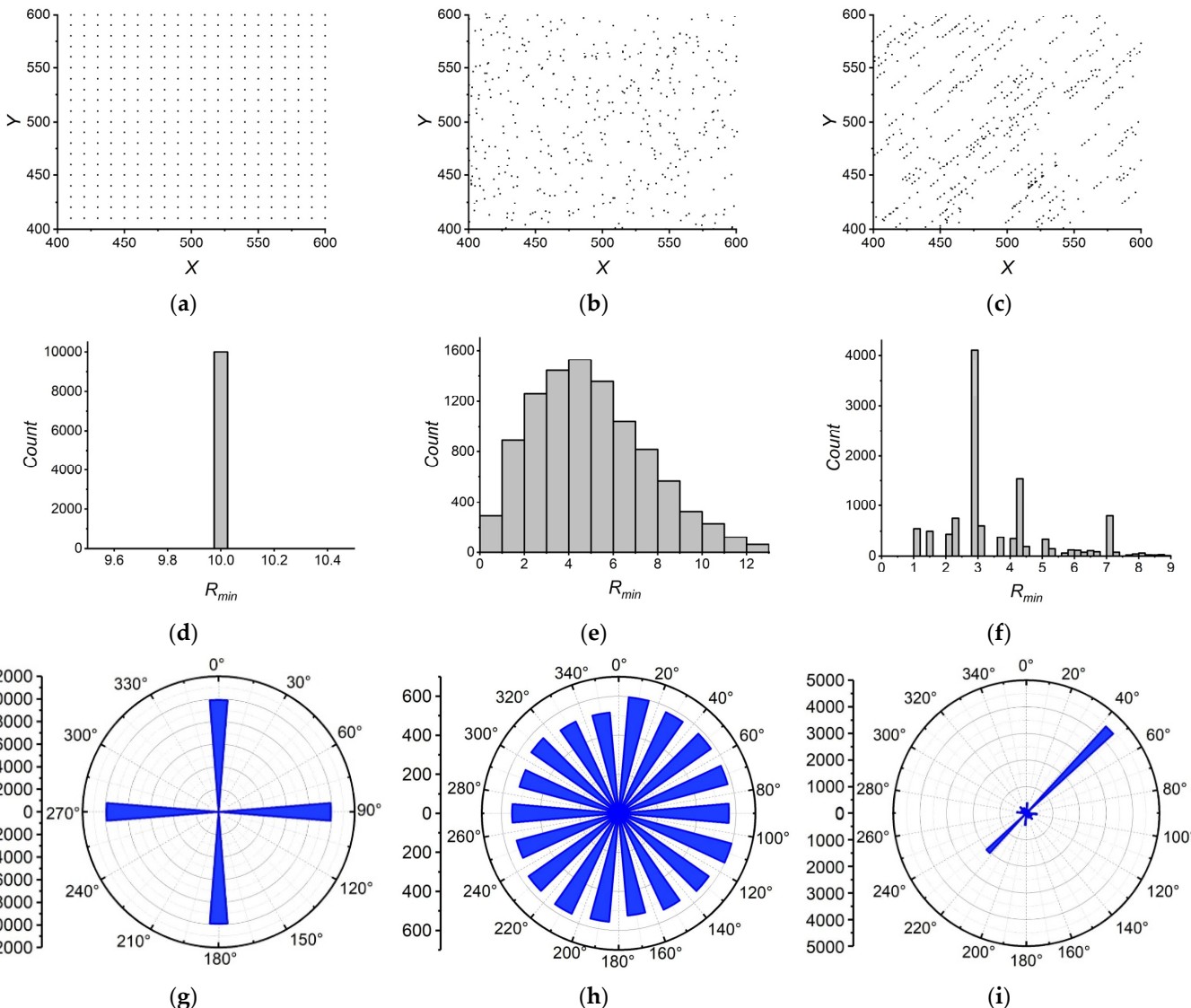

**Figure 2.** Model pores spatial distributions: fully ordered in a squared net (**a**), random homogeneous (**b**), random with a preferred orientation of the nearest pores pairs (**c**). Corresponding distributions of the minimum distances $P_{Rmin}$ between the nearest pores (**d–f**) and orientation diagram $P\varphi$ in polar coordinates—the distributions of the values of the inclination angles of the segments between the centers of the nearest pores (**g–i**).

To investigate these parameters and distributions in porous films with known parameters, we performed some model calculations. The above-mentioned distributions were found for three different model cases (pore spatial distribution patterns):

(a) a strongly ordered spatial distribution, when the pores were located in the nodes of the square grid with a certain period *R*;

(b) a random homogeneous distribution with a certain surface density *ρ*;

(c) a distribution with the preferred orientation distribution of the pores, i.e., "anisotropy" (here, the nearest pores were oriented with respect to each other according to a preferred angle *φ*) and with three selected preferable minimal distances between nearest pores (~3, 4, and 7 a.u.).

For all three cases, the size of the model sample surface was 1000 a.u. per 1000 a.u, and the number of generated pores was 10,000, with a diameter of 1 a.u. For each case, the histograms of the distances distribution $P_{Rmin}$ between the nearest pairs of pores and the histogram of the orientations of the nearest pores pairs $P\varphi$ were studied (see Figure 2).

As we see in Figure 2, for a fully ordered case (pores in nodes of the squared net), the distribution of distances between nearest pores pairs is the delta function (Figure 2d), and the orientation diagram has only four petals at angles of 0, 90, 180 and 270° (Figure 2g). For the random homogeneous distribution, there was a (quasi) normal distribution $P_{Rmin}$ (Figure 2e) and a ring histogram $P\varphi$ without any preferred orientation (Figure 2h).

Finally, in the case of the "anisotropic" distribution with the preferred orientation of the pores (nearest pores oriented with respect to each other according to a preferred angle *φ*) and with three selected preferable minimal distances between nearest pores we obtained a strongly spiked histogram $P_{Rmin}$ (Figure 2f), with clearly resolved maxima at ~2.9, 4.3, and 7.1 a.u, that fully corresponded to the modeling parameters. The ring histogram $P\varphi$ showed preferred orientations of the nearest pores pairs at 45 and 225° (Figure 2h), which were chosen for the simulations.

## 5. Experimental Results and Discussion

When studying samples by the AFM method, it is necessary to select the optimal scanning modes to avoid the appearance of various artifacts (blurring, noise) (Figure 3). As the scanning statistics show, such artifacts arise when the feedback coefficient is incorrectly selected, the probe is too close to the samples, and the scanning speed is too high.

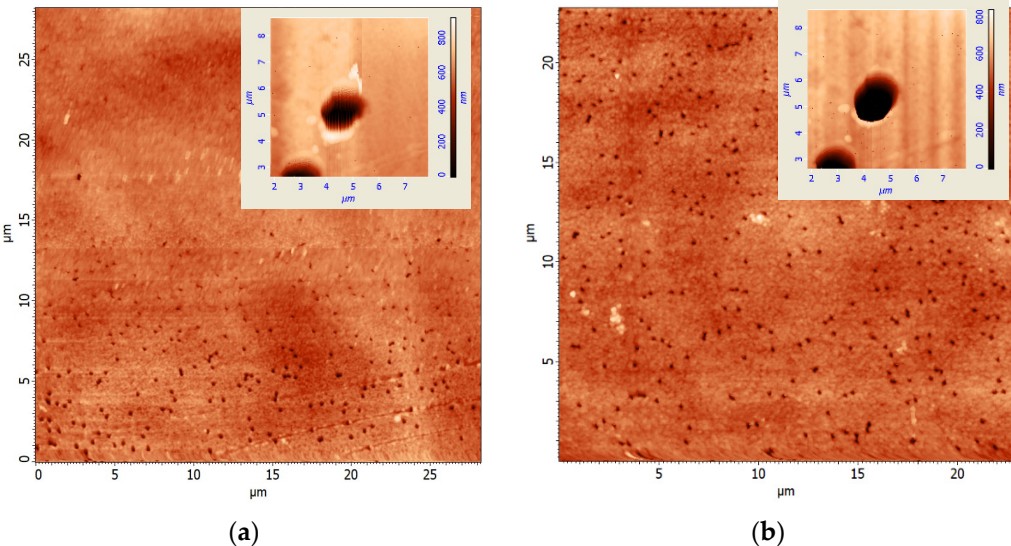

| (a) | (b) |
|---|---|

**Figure 3.** AFM images of the same section of the TM and of a single pore (inset) with non-optimal scanning parameters (**a**) and with optimal scanning parameters (**b**).

To obtain information on the distributions of the pores parameters, the scanning scale of 20 × 20 μm was used, with a step size of ~78 nm (resolution 256 × 256 points) and a scanning frequency of 0.5 Hz. The optimum scanning parameters were found to avoid undesirable artifacts in AFM images like blurring and noise (see Figure 3a). When the parameters of single pores were investigated, the measurements were carried out using a smaller scan scale (from 5 to 10 μm) with the resolution of 512 × 512 points, with the same scanning frequency.

To determine the positions of the centers $\{X_c, Y_c\}$ and diameters $D$ of the pores on the AFM images, we used a special software, which works in a semi-automatic mode by approximation of measured pores profiles with 2D-Gaussian (Figure 4a).

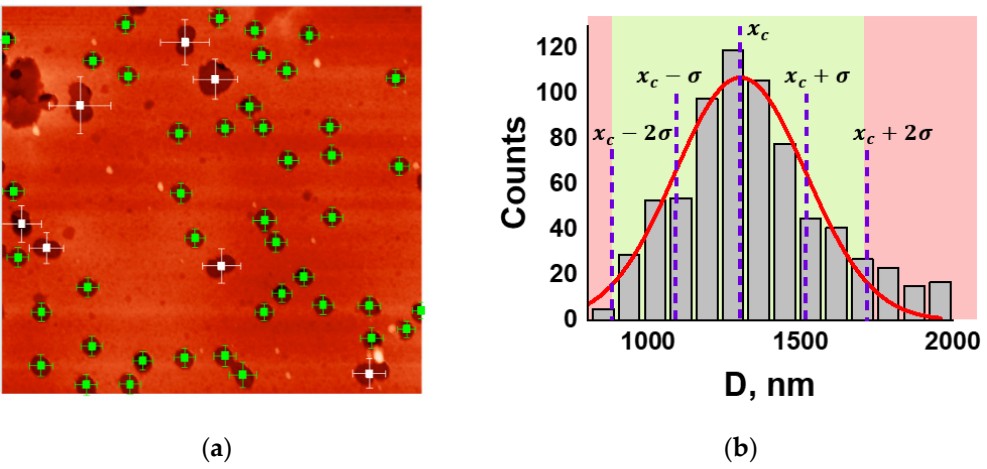

(**a**)                    (**b**)

**Figure 4.** AFM image of the surface of a PETP TM with marked positions (squares) and calculated pore sizes (dashes) (**a**). Histogram of pore diameters distribution, where the green zone in the center denotes the area of diameters $D$ for "good" pores, and the red zones at the edges indicate defective pores (**b**). The red line shows the result of approximating the obtained distribution by the Gaussian function (with standard deviation of $\sigma = 215$ nm).

Figure 4b shows the distribution of the measured pore diameters $P_D$ and marks the range of "true" pore diameters (green area in the center), i.e., pores that are not considered defective according to their deviation from the mean value.

Moreover, this criterion made it possible to exclude defective pores if the distance between their centers was comparable to their sizes. Such pores are indicated in Figure 4a by the white square in their centers.

The results obtained for PP TM are shown in Figure 5a–d. The images of 1951 pores were analyzed. The average value of the pore diameter was approximately 182.6 nm (at RMS about 66 nm). We determined that 16% of the pores were defective. Further, the data for the remaining 1641 pores were analyzed. The average value of the minimum distance between PP pores was approximately 697.9 nm. The average pore density was around $0.6 \times 10^8$ pores/cm$^2$. Similar measurements were performed for PETP- and PC-based TM. The obtained pore parameters are shown in Table 1. The results obtained for PETP are shown in Figure 5e–h, and those for PC are shown in Figure 5i–l.

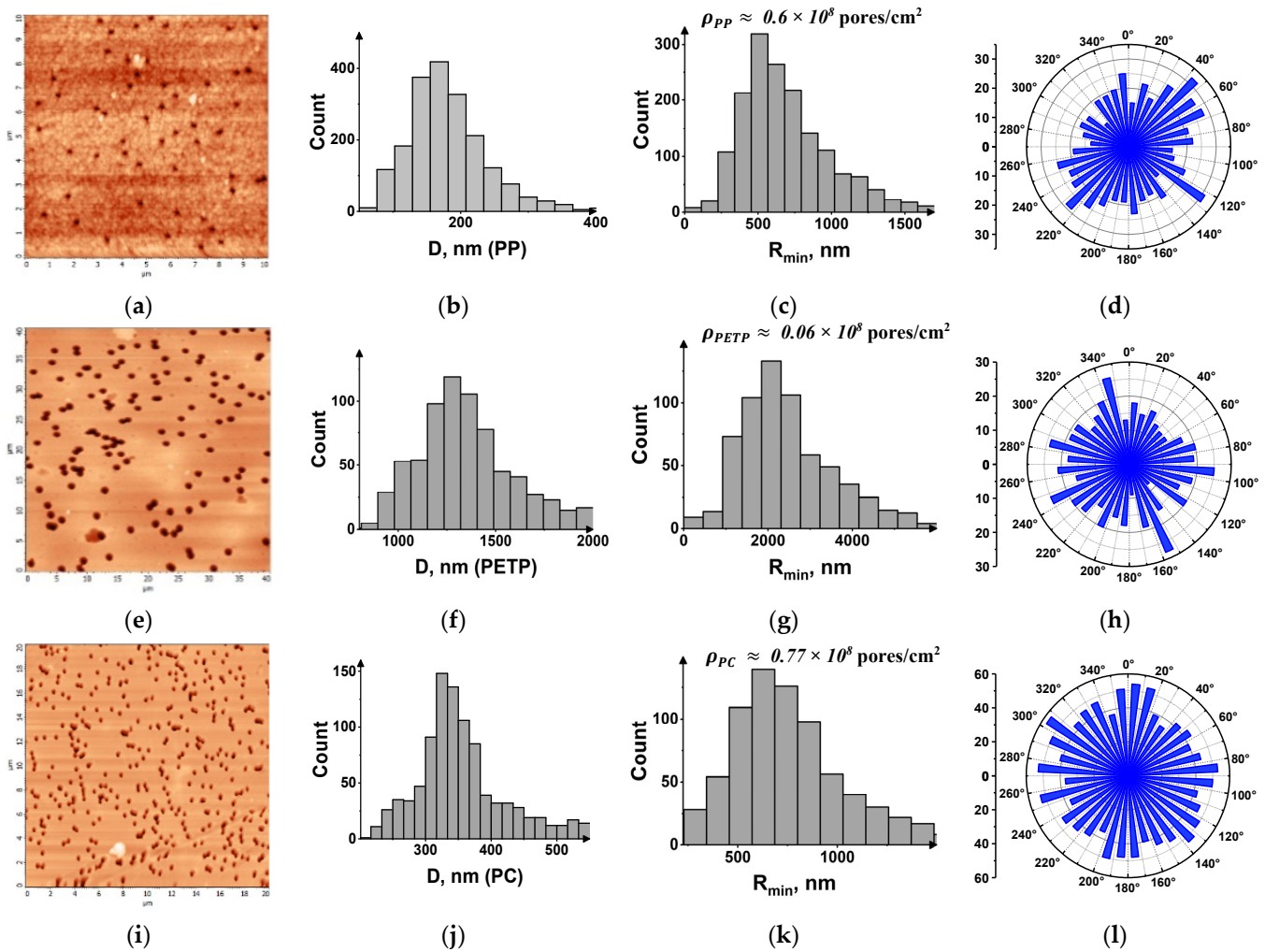

**Figure 5.** Obtained results for different samples. PP TM: AFM image (**a**), histogram of the diameters distribution (**b**), histogram of the distribution of the minimum distances $P_{Rmin}$ between the nearest pores (**c**), orientation diagram $P\varphi$ in polar coordinates—the distributions of the values of the inclination angles of the segments between the centers of the nearest pores (**d**). The same for PETP TM (**e**–**h**) and PC TM (**i**–**l**).

**Table 1.** Pore parameters for three TM samples.

| TM Material | PP | PETP | PC |
|---|---|---|---|
| Total number of pores | 1951 | 771 | 1014 |
| Average pore diameter $<D>$, nm | 183 | 1430 | 375 |
| RMS $<D>$, nm | 66 | 415 | 112 |
| Relative number of defective pores, $\eta_{defect}$, % | 16 | 16 | 26 |
| Average value of the minimum distance between pores $<R_{min}>$, nm | 698 | 2514 | 770 |
| Average pore density $\rho$, $\times10^8$ pores/cm$^2$ | 0.6 | 0.06 | 0.77 |

The analysis of the AFM images of the TM samples using the developed procedure made it possible to conclude that the pore orientation had no preferred direction and to obtain the characteristic values of pore density.

We developed a technique for processing AFM images which can also be used for processing images obtained by the SEM method. An important advantage of the AFM technique over the SEM method is the ability to obtain three-dimensional data and the

profiles of pore edges for the precise determination of the coordinates of their centers and diameters, which is of particular interest in the study of track membranes.

## 6. Conclusions

In this paper, a technique for the structural characterization of porous polymer films (track-etched membranes) was implemented by using atomic force microscopy (AFM) and subsequent computer analysis of the data. The technique allows obtaining the distribution of the diameters of the pores (and identifying defective pores), the minimum distances between the nearest pores, and their average spatial density. We suggested a method for the characterization of surface anisotropy in pores distribution by analyzing the orientation of the nearest pores pairs. The main criteria for defective pores selection were their anomalous large diameters and the distance between the nearest pores.

Three model pore distributions in TM were realized and investigated by numerical simulation (uniform distribution, random distribution, and spatially oriented distribution). In all cases, the distributions of the minimal distances between the nearest pores were analyzed, as well as ring histograms which demonstrated the distribution of the orientation of the nearest pores pairs. In the case of a random distribution with a preferred direction, the orientation ring histogram had peaks that demonstrated the formation of some "chains" and "agglomerates" of pores. It means that this way of statistical processing of AFM data allows establishing the presence (or absence) of the preferred direction of pores distribution in a sample.

The analysis of the images of the three types of TM (based on PP, PETP, and PC) made it possible to conclude that there was no surface anisotropy (a preferred direction of pores orientation) in all studied samples.

From the analysis of the experimental data, it can be concluded that the average value of pore density in the PETP sample markedly differed from that of the other two samples by more than an order of magnitude. Apparently, this was because the minimum distance between the nearest pores in this sample was ~2 μm, which was three times greater than that in the other two samples; in addition, the average pores diameter (1430 nm) was very large (183 and 375 nm). The developed method allows a more complete description of the parameters of porous membranes, including the coordinates of the centers, the diameters, the minimum distance, the density, and the direction of pore distributions.

**Author Contributions:** Conceptualization, A.V.G. and I.Y.E.; Data curation, A.I.A. and A.V.N.; Formal analysis, A.V.G.; Investigation, A.V.G., M.A.D. and A.I.A.; Methodology, A.I.A.; Project administration, A.V.N.; Resources, I.Y.E.; Software, A.V.G.; Supervision, A.V.N.; Validation, I.Y.E.; Visualization, A.V.G. and K.R.K.; Writing—original draft, A.V.G., M.A.D. and A.I.A.; Writing—review & editing, K.R.K., I.Y.E. and A.V.N. All authors have read and agreed to the published version of the manuscript.

**Funding:** The work was supported by the Russian Foundation for Basic Research (project No. 20-02-00871 realized at the Institute of spectroscopy RAS) and the Ministry of Education of Russia (State Contract at MPGU AAAA-A20-120061890084-9 for Laboratory of Physics of Advanced Materials and Nanostructures). The authors are members of the leading scientific school of Russia "Optical-spectral nanoscopy of quantum objects and diagnostics of promising materials" (project NSh-776.2022.1.2).

**Institutional Review Board Statement:** Not applicable.

**Informed Consent Statement:** Not applicable.

**Acknowledgments:** The authors acknowledge Pavel Apel (JINR, Dubna) for the samples of etched tracking membranes [2,15].

**Conflicts of Interest:** The authors declare that they have no conflict of interest.

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
