# Peer review of "AFM Characterization of Track-Etched Membranes: Pores Parameters Distribution and Disorder Factor"

_applsci, doi:10.3390/app12031334_

Round 1

Reviewer 1 Report

A revision of the test is required. The authors sometimes are not clear enough and the methods and results do not benefit from the present description.

For instance:

  1. The section describing the sample preparation is not clear. First, it seems they made the samples starting from purchased films but then they mention ‘a manufacturer’ which remains in the end unknown.
  2. Line 166: this sentence is not intelligible.
  3. 'surface topography maps' is redundant. I would remove 'surface'.

Other issues:

  1. NC01 is also made by NT-MDT, specify.
  2. Line 93: I doubt the full amplitude is only 2nm. Typically is much more.
  3. 'Tomograms'  is rarely appropriate for AFM maps. Definitely, this is not the case.
  4. In Figure 2, the axis labels are not of the same size.
  5. In Table 1, the digits reported with a comma do not make sense as the error bars are much larger. Round off to the last digit.

Author Response

First of all, we thank the reviewer for the fruitful remarks. On their base, we rewritten several parts of the manuscript, improve some bags and misprints.

A revision of the test is required. The authors sometimes are not clear enough and the methods and results do not benefit from the present description.

Answer: We have added corresponding statements in the text and tried to improve the text quality to make it clearer. All the improvements are marked by yellow highlight.

For instance:

The section describing the sample preparation is not clear. First, it seems they made the samples starting from purchased films but then they mention ‘a manufacturer’ which remains in the end unknown.

Answer: For our study, we have used ready-made films from the manufacturer - Flerov Laboratory of Nuclear Reactions of the Joint Institute for Nuclear Research (Dubna). We have added corresponding clarification in the text. These samples have been developed by the group of Prof. Pavel Apel who is a world-wide recognized leader in the field of physics of track-etched polymer membranes. We add references links and acknowledgments.

Line 166: this sentence is not intelligible.

Answer: We have excluded this part of the text, remaining from the previously deleted section.

'surface topography maps' is redundant. I would remove 'surface'.

Answer: We have corrected this issue in the text.

Other issues:

NC01 is also made by NT-MDT, specify.

Answer: We have corrected this issue.

Line 93: I doubt the full amplitude is only 2nm. Typically is much more.

Answer: Indeed, the typical values of probe oscillations are greater than 10 nm for AFM measurements in the semicontact method. These values are needed to ensure that the energy in the lever arm is much higher than the energy lost in each cycle when hitting the sample surface, thus avoiding sticking the probe tip to the sample surface.

To study the surface of our samples of polymer films, an amplitude value of about 2 nm or more was sufficient, while we did not observe any artifacts or coordinate shifts in our results.

'Tomograms'  is rarely appropriate for AFM maps. Definitely, this is not the case.

Answer: We have corrected this issue.

In Figure 2, the axis labels are not of the same size.

Answer: We have corrected this issue. Also, we have tried to improve the quality of all figures.

In Table 1, the digits reported with a comma do not make sense as the error bars are much larger. Round off to the last digit.

Answer: We have corrected this issue.

In general, we improved a few parts of the manuscript to make it more understandable for readers. All the changes we mark by the yellow highlights.

Reviewer 2 Report

Reviewer report on the manuscript AFM characterisation of Track Etched Membranes: Pores parameters Distribution and Disorder Factor“ by A.V. Golovanova et al.

The paper describes use of AFM technique for characterization of etched ion tracks in polycarbonate, polypropylene and polyethylene terephthalate. The characterization has been automated by developed code for analysis of the AFM images. The paper is well written, and illustrates well how the code works.

The analysis of AFM images has always been a bit problematic, so the results presented here are clearly worthy of publishing in Appl. Sci. English language is good, but I would still propose to check it once more by the authors.

Finally, I would advise authors to clarify to the readers that measured instrumental function of device (obtained on the micron-sized sample) is adequate for the analysis of track etched pores that have diameter size close to 200-300 nm (at least for presented results of PP and PC track membranes).

Author Response

First of all, we thank the reviewer for the fruitful remarks. On their base we rewrite several parts of the manuscript, improve some bags and misprints.

English language is good, but I would still propose to check it once more by the authors.

Answer: In the revised version we tried to make up for this flaw.

Finally, I would advise authors to clarify to the readers that measured instrumental function of device (obtained on the micron-sized sample) is adequate for the analysis of track etched pores that have diameter size close to 200-300 nm (at least for presented results of PP and PC track membranes).

Answer: We thank the referee for this important note. We improve this part of the paper strongly. In particular, we find some bags (see Eq. 1) in the procedure of analysis of the instrumental function, and describe this analysis in more detail. So, with correct convolution (Eq. 1) the width of the Gauss function was found as high as 40 nm, and this value is defined by the shape of the AFM tip. The real precision of pore edge recognition is much higher and we estimate it on the level of 5 nm. We rewrite this part of the manuscript.

In general, we improved a few parts of the manuscript to make it more understandable for readers. All the changes we mark by yellow highlights.